# Severity of Respiratory Viral Diseases and the Impacts of Underlying Medical Conditions During the Omicron Subvariant Dominant Epidemics—A Comparative Study of SARS-CoV-2, Influenza Virus and Respiratory Syncytial Virus

**DOI:** 10.3390/pathogens14060543

**Published:** 2025-05-29

**Authors:** Yu Jung Choi, Joon Young Song, Seong-Heon Wie, Won Suk Choi, Jacob Lee, Jin-Soo Lee, Young Keun Kim, Shin Woo Kim, Sun Hee Lee, Kyung-Hwa Park, Hye Won Jeong, Jin Gu Yoon, Hye Seong, Eliel Nham, Ji Yun Noh, Hee Jin Cheong, Woo Joo Kim

**Affiliations:** 1Division of Infectious Diseases, Department of Internal Medicine, Guro Hospital, Korea University College of Medicine, Seoul 08308, Republic of Korea; 2Vaccine Innovation Center-KU Medicine (VIC-K), Seoul 02841, Republic of Korea; 3Division of Infectious Diseases, Department of Internal Medicine, St. Vincent’s Hospital, College of Medicine, The Catholic University of Korea, Suwon 06591, Republic of Korea; 4Division of Infectious Diseases, Department of Internal Medicine, Ansan Hospital, Korea University College of Medicine, Ansan 02841, Republic of Korea; 5Division of Infectious Diseases, Department of Internal Medicine, Kangnam Sacred Heart Hospital, Hallym University College of Medicine, Seoul 05355, Republic of Korea; 6Division of Infectious Diseases, Department of Internal Medicine, Inha University School of Medicine, Incheon 22332, Republic of Korea; 7Division of Infectious Diseases, Department of Internal Medicine, Yonsei University, Wonju College of Medicine, Wonju 26426, Republic of Korea; 8Division of Infectious Diseases, Department of Internal Medicine, Kyungpook National University School of Medicine, Daegu 41404, Republic of Korea; 9Division of Infectious Diseases, Department of Internal Medicine, Pusan National University School of Medicine, Pusan 50612, Republic of Korea; 10Division of Infectious Diseases, Department of Internal Medicine, Chonnam National University Medical School, Gwangju 61469, Republic of Korea; 11Department of Internal Medicine, Chungbuk National University College of Medicine, Cheongju 28644, Republic of Korea

**Keywords:** SARS-CoV-2, influenza, respiratory syncytial virus, pneumonia, comorbidity

## Abstract

After the transition of coronavirus disease 2019 (COVID-19) from a pandemic to an endemic phase, data on respiratory viral infections remain limited. This study compared the clinical outcomes of SARS-CoV-2, influenza virus (INFV), and respiratory syncytial virus (RSV) infections and investigated how underlying medical conditions influence disease severity. During Omicron subvariant dominant periods, we conducted a multicenter, retrospective cohort study including laboratory-confirmed cases of SARS-CoV-2, INFV, and RSV infections in hospitalized patients aged ≥ 19 years. We compared demographic characteristics and clinical outcomes and analyzed the association between underlying comorbidities and severity of infection. A total of 1850 cases with SARS-CoV-2, 98 with INFV, and 63 with RSV infections were analyzed. Notable differences in the occurrence of fever, cough, sputum, and dyspnea were observed among patients with the three different viral infections. Pneumonia was diagnosed more frequently in patients with RSV infection (65.6%) compared to those with INFV infection (42.9%) and SARS-CoV-2 (34.4%) (*p* < 0.01). For patients with SARS-CoV-2 infection, the risk of pneumonia increased by 47% in the moderate-risk group and 37% in the high-risk group. Among hospitalized patients, pneumonia was more frequently identified in patients with RSV infection, with statistical significance. Furthermore, the presence of medical conditions significantly increased the risk of developing pneumonia.

## 1. Introduction

The coronavirus disease 2019 (COVID-19) pandemic significantly impacted the landscape of respiratory viral infections [1,2,3,4,5]. Reinforced mask use and social distancing effectively suppressed the circulation of other respiratory viruses. Notably, there was no influenza epidemic for two and a half years during the COVID-19 pandemic.

On 5 May 2023, the World Health Organization declared the end of the COVID-19 public health emergency, marking the transition to an endemic era. With the easing of social distancing measures, seasonal respiratory virus epidemics have resumed. Influenza virus (INFV), respiratory syncytial virus (RSV), and severe acute respiratory syndrome coronavirus 2 (SARS-CoV-2) are circulating concurrently, leading to high morbidity and mortality. Actually, in the Republic of Korea, for instance, the seasonal influenza epidemic began earlier than usual in September 2022 [6].

Most respiratory viruses can cause lower respiratory tract infections, including bronchiolitis and pneumonia, leading to hospitalization and death. A systematic review of 31 observational studies on community-acquired pneumonia (CAP) discovered that 24.5% of cases were caused by viral infections, with approximately half (10%) being viral/bacterial co-infections [7]. Notably, older age, alcoholism, and underlying medical conditions are established risk factors for severe pneumonia requiring hospitalization [8,9,10]. Additionally, studies have revealed that the average direct medical cost for domestic hospitalized patients with CAP is more than double in the high-risk group ($2841 per person) compared to the low-risk group ($1263) [11].

SARS-CoV-2, INFV and RSV are the leading causes of respiratory viral infections in adults and frequently cause pneumonia. However, the risk factors for pneumonia and severe complications may differ among these viruses. This study aimed to compare the clinical outcomes and risk factors of severe infections among these three representative respiratory viral infections (SARS-CoV-2, INFV and RSV).

## 2. Materials and Methods

### 2.1. Study Design and Data Sources

We conducted a multicenter, retrospective cohort study involving eight hospitals located in various regions (Seoul, Suwon, Ansan, Incheon, Wonju, Daegu, Pusan, Gwangju, and Chungbuk) of the Republic of Korea, all participating in the Hospital-based Influenza Morbidity and Mortality surveillance system (HIMM). This study was conducted between 1 September 2022, and 31 August 2023, during periods when the SARS-CoV-2 Omicron subvariants circulated dominantly. We included all hospitalized patients aged 19 years or older with laboratory-confirmed SARS-CoV-2, INFV, or RSV infections during this period. Patients younger than 19 years were excluded. Respiratory virus testing was performed as part of standard clinical diagnostic procedures.

Clinical and laboratory information was obtained from the electronic medical records of each participating hospital. We analyzed and compared demographic characteristics and clinical outcomes (pneumonia, intensive care unit [ICU] admission, acute respiratory distress syndrome [ARDS], and mortality) across the three different viral groups. Additionally, we evaluated the risk of severe infection based on the presence of underlying chronic medical conditions. Vaccination status was verified through patient history (obtained through one-on-one interviews), hospital electronic records, and the Korea Disease Control and Prevention Agency’s vaccination database.

### 2.2. Definition

Each viral infection was defined as a case meeting the following criteria: for SARS-CoV-2 infection, a positive test result on either a SARS-CoV-2 rapid antigen test (RAT) or a SARS-CoV-2 reverse transcription-polymerase chain reaction (RT-PCR) test; for INFV infection, a positive test result on either an INFV RAT or a respiratory virus PCR test; and for RSV infection, a positive respiratory virus PCR test. INFV and RSV were tested using multiplex PCR for respiratory viruses. We defined individuals as vaccinated if they had received the vaccine at least 14 days prior to the diagnosis.

Patients were classified into risk groups according to their underlying medical conditions. Individuals with no underlying medical conditions were assigned to the low-risk group. The moderate-risk group included those with at least one of the following medical conditions: diabetes mellitus, cardiovascular disease, cerebrovascular disease, neuromuscular disease, chronic obstructive pulmonary disease, asthma, other chronic lung diseases, chronic renal disease, or chronic liver disease. Finally, the high-risk group included patients with at least one of the following immunocompromising conditions: splenic dysfunction (including post-splenectomy), hematologic malignancy, solid cancer undergoing chemotherapy, solid organ transplantation, stem cell transplantation, human immunodeficiency virus infection, or high-dose corticosteroid use (≥20 mg/day of prednisone or equivalent for two or more weeks).

### 2.3. Statistical Analysis

Statistical analyses were carried out using IBM SPSS Software version 25.0 (IBM Corp., New York, NY, USA). To compare clinical features and outcomes among the three virus groups, categorical variables were assessed using the Chi-square test, while continuous variables were examined with analysis of variance (ANOVA). Categorical variables, such as the presence or absence of symptoms, were analyzed and expressed as percentages (%). Quantitative variables, such as age or duration of hospitalization, were expressed as mean ± standard deviation, rounded to one decimal place. To control for false positives, post hoc analyses were conducted following significant findings. For chi-square tests, adjusted residuals were used to identify cell-level differences, and for ANOVA, Bonferroni correction was applied. A two-tailed *p*-value of less than 0.05 was considered statistically significant.

To evaluate the association between underlying medical conditions and the development of clinical outcomes (pneumonia, ICU admission, and mortality), logistic regression analysis was used. This analysis estimated the odds ratio (OR) for each outcome based on the presence of specific conditions. Age, sex, and vaccination status were included as covariates in the model to account for their potential influence. Separate analyses were conducted for each underlying disease and risk group (low, moderate, and high).

### 2.4. Ethics Statement

This retrospective study received approval from the Institutional Review Board of each participating hospital: Korea University Guro Hospital (2022GR0360), Korea University Anam Hospital (2022AN0449), Korea University Ansan Hospital (2022AS0226), St. Vincent’s Hospital (VC22TIDI0150), Kangnam Sacred Heart Hospital (2022-07-016), Inha University Hospital (2022-07-036), Chungbuk National University Hospital (2022-08-022), and Gil Medical Center (GAIRB2022-306). This study was conducted in line with the ethical principles outlined in the Declaration of Helsinki by the World Medical Association. Given the retrospective study design, the requirement for written informed consent was waived.

## 3. Results

### 3.1. Clinical Characteristics and Outcomes of SARS-CoV-2, INFV, and RSV

A total of 2011 hospitalized patients were included in the analysis across eight medical centers from 1 September 2022, to 31 August 2023, during periods when the SARS-CoV-2 Omicron subvariants circulated dominantly: 1850 with SARS-CoV-2, 98 with INFV, and 63 with RSV infection (Table 1).

The mean age of the patients ranged from 68 to 70 years, with over 65% of patients in all three groups being older than 65 years. No significant differences in underlying medical conditions were observed among the groups, although chronic lung disease and a history of bone marrow transplantation were both more prevalent in RSV patients.

Clinical presentations varied significantly among the viral groups (Table 1). Patients with SARS-CoV-2 infection were prone to be asymptomatic and experienced less frequent cough and sputum compared to those with INFV or RSV infection. Among patients positive for each virus, those with INFV infection exhibited a higher prevalence of fever (76/98, 77.6%, *p* < 0.001), whereas dyspnea was more common among patients with RSV infection (40/63, 63.5%, *p* < 0.001; Table 1). Regarding laboratory findings, the RSV group demonstrated significantly higher erythrocyte sedimentation rates (ESR) and C-reactive protein (CRP) levels compared to the other groups. No other significant differences were observed in the laboratory findings. Chest radiography revealed pneumonia more frequently in the RSV group (65.6%) compared to the other groups (42.9% and 34.4% in the INFV and SARS-CoV-2 groups, respectively; *p* < 0.01). There was no difference in vaccination status between the groups.

The incidence of pneumonia differed markedly across the three viral groups: 39.8%, 24.8%, and 68.8% in patients with SARS-CoV-2, INFV, and RSV, respectively (*p* < 0.001; Table 2). Other clinical outcomes, such as ICU admission, ARDS, mortality, and the duration of hospitalization, did not show significant differences among the three groups. Both adjusted residuals analysis (using a significance threshold of ±1.96) and Bonferroni correction consistently confirmed the presence of significant cell-level differences across all relevant comparisons.

### 3.2. Impact of Underlying Medical Conditions on Disease Severity

We analyzed the risk of pneumonia development based on underlying medical conditions (Table 3).

Compared to the low-risk group, the incidence rate of SARS-CoV-2 pneumonia was significantly higher in the moderate-risk group (1.47-fold increase; 95% confidence intervals [CI]: 1.07–2.01, *p* = 0.017) and showed a trend toward being higher in the high-risk group (1.37-fold increase; 95% CI: 0.90–1.93, *p* = 0.074) (Table 3). Within the moderate-risk group, the incidence of pneumonia increased further with a rising number of chronic medical conditions (from 1.46 to 1.54 compared to the low-risk group). Details on the risk of SARS-CoV-2 pneumonia associated with specific chronic conditions are provided in Appendix A. INFV and RSV pneumonia followed a similar trend, although the increases in risk were not statistically significant.

The risk of ICU admission was also evaluated based on underlying conditions (Table 4).

For patients with SARS-CoV-2 infection, the risk of ICU admission in the moderate-risk group with three or more chronic conditions was significantly higher (a 1.38-fold increase; 95% CI: 0.96–1.99, *p* = 0.001) compared to the low-risk group. Notably, the risk of ICU admission in RSV patients with three or more chronic conditions was even greater (28.23-fold increase; 95% CI: 1.38–576.83, *p* = 0.030). Interestingly, among patients with SARS-CoV-2 infection, the ICU admission rates were the highest in the moderate-risk group, followed by the low-risk and high-risk groups, though these differences did not reach statistically significant. No significant differences were observed among risk groups among patients with INFV infection.

While not statistically significant, the mortality rate was consistently higher in the high-risk group compared to the moderate and low-risk groups for all three viral infections (Table 5). Additionally, within the moderate-risk group, there was a trend toward increasing mortality rates with a higher number of chronic medical conditions.

## 4. Discussion

This study evaluated the clinical outcomes of three major respiratory viruses (SARS-CoV-2, INFV, and RSV) and assessed whether underlying medical conditions influenced disease severity. The clinical presentations differed markedly; patients with SARS-CoV-2 infection exhibited a higher rate of asymptomatic infection and those with INFV infection were prone to experience fever. In comparison, patients with RSV infection had a significantly higher hospitalization rate due to dyspnea. Notably, pneumonia was significantly more prevalent among hospitalized patients with RSV infections. In patients with SARS-CoV-2 infection, the risk of pneumonia increased with the number of underlying medical conditions (1.47-fold increase in the moderate-risk group and 1.37-fold increase in the high-risk group). This trend was similar to that of the ICU admission rates. There was no significant difference in mortality rates among the three viral groups, which might be due to the high proportion (approximately 70%) of adults aged over 65 years in our study population.

Patients with RSV infections presented with severe symptoms, including dyspnea, elevated ESRs and CRP levels, and frequent pneumonia observed on chest X-rays (Table 1). However, a previous prospective study on CAP discovered that only 3.7% of adults with radiologically confirmed CAP tested positive for RSV [12]. This suggests that RSV infections might be underdiagnosed in clinical practice. Among the older population, RSV contributes to a significant disease burden, potentially equal to or exceeding that of INFV infection [13,14]. Unlike SARS-CoV-2 and INFV infections, RSV RAT is not commonly used in routine clinical practice due to the lack of effective antiviral agents. Consequently, diagnosis often relies on respiratory viral PCR testing, particularly in cases of suspected pneumonia. This may lead to an underestimation of the real disease burden and overestimation of the severity for RSV infection, highlighting the need for further well-designed prospective studies.

Several studies have investigated factors influencing the severity of respiratory virus infections [15,16,17,18,19,20,21]. Large-scale prospective studies and systematic reviews have identified old age, male sex, heart failure, and chronic kidney disease as risk factors for severe COVID-19, with age being the strongest predictor of severe illness and mortality [15,16]. Similarly, Federico et al. reported a significant increase in the burden of RSV and INFV infections among the older population [17]. As for RSV infection, Edward et al. reported a 3.97-fold increased risk of hospitalization in patients with underlying lung disease [18]. Additionally, a retrospective study conducted in China reported lower survival rates following RSV infection among elderly patients and those with congestive heart failure or chronic obstructive pulmonary disease [22]. Similarly, a study from Croatia found that chronic conditions, particularly cardiovascular and respiratory diseases, were strongly associated with higher complication rates in RSV-infected patients [23]. Consistent with these findings, our study identified old age and chronic medical conditions (both moderate and high-risk groups) as risk factors for pneumonia.

However, compared to a previous study where the risk of CAP increased by 4.32–7.57 fold in the moderate-risk group compared to the low-risk group (healthy individuals), the effect of chronic medical conditions appeared to be lower in our study [19]. This difference is potentially attributable to our study population. Unlike previous studies that included younger and outpatient populations, ours focused on hospitalized patients, most of whom were older adults. In the Republic of Korea, over 90% of individuals aged 65 years and older have at least one chronic medical condition, and more than half have three or more [24,25,26]. This high prevalence of underlying medical conditions within our study population, composed primarily of older adults, may explain the seemingly limited effect of these conditions on disease severity. For instance, our study found a 2.37-fold increased risk of pneumonia for asthma and a 1.79-fold increased risk for other chronic lung diseases (Appendix A), whereas previous studies reported a 4.16-fold increased risk in patients with chronic lung diseases. These findings suggest that the effect of underlying medical conditions on disease severity may be less pronounced in older adults.

ICU admission rates for SARS-CoV-2 were higher in the moderate-risk group and relatively lower in the high-risk group. This could be due to the larger proportion of high-risk patients with solid cancer or hematologic conditions, who might have chosen less aggressive treatments and declined ICU admission (Appendix A). Interestingly, age did not significantly influence ICU admission rates, suggesting that suggesting that older patients may have led to the avoidance of unnecessary resource allocation Similarly, ICU admission rates decreased in patients with solid cancers. However, the high-risk group exhibited a trend toward higher mortality rates, which was also observed in older patients with solid cancers (Appendix A).

This study had some limitations. First, the retrospective design may have led to an underestimation of the disease burden associated with RSV infections, as mild cases could have been missed or underreported. In particular, mild RSV infections in adults are often underestimated, underscoring the need for future research to develop and validate a standardized case definition that can facilitate more accurate diagnosis and monitoring, especially in outpatient and community settings. Large-scale, prospective studies are needed to definitively assess whether ICU admission or mortality rates are higher among patients with RSV infection compared to those with SARS-CoV-2 or INFV infections. Secondly, this study was limited to a single season shortly following the COVID-19 pandemic. The limited timeframe and relatively few INFV and RSV cases reduced the statistical power of certain comparisons. Consequently, interpretations of incidence and clinical outcomes across SARS-CoV-2, INFV, and RSV should be approached with caution, particularly given the disproportionately low number of RSV cases, which may compromise the robustness of the findings. Third, as mentioned above, most of our study population consisted of elderly hospitalized patients (69.4% aged ≥ 65 years), which may limit the generalizability of our findings to other populations. In contrast to the inconsequential impact of RSV in adults, RSV is well-known as the leading cause of bronchiolitis and pneumonia in children. Although a significant increase was observed in bronchiolitis by non-RSV pathogens (INFV A/B, metapneumovirus, adenovirus) following the COVID-19 pandemic, RSV still accounted for the largest proportion (40%) of the cases [27,28]. Finally, this study did not explore inter-viral interactions, and future research should investigate co-infections and biomarkers to better understand disease severity and prognosis.

Current RSV surveillance systems are methodologically limited, as RSV diagnosis is primarily based on PCR testing conducted on hospitalized patients or those visiting emergency departments; rapid diagnostic tests are generally not performed in outpatient clinics. However, despite these limitations, our study contributes preliminary insights that warrant further investigation using more comprehensive and representative datasets. Given the currently low level of awareness among clinicians regarding RSV infection, it is necessary to conduct research to assess their knowledge and awareness of the disease. In addition, a large-scale prospective study using highly sensitive PCR-based testing in patients presenting with respiratory or febrile symptoms is needed to evaluate the clinical characteristics and disease burden of RSV infection by age group and comorbidity status. Furthermore, studies assessing the duration of neutralizing antibody immunity and the rate of reinfection following RSV infection are essential for establishing future RSV vaccination strategies.

This study highlights that SARS-CoV-2 is more severe in patients aged 65 and older, particularly those with multiple underlying medical conditions, emphasizing the critical role of vaccination in these populations. Additionally, the potential underestimation of the RSV disease burden suggests a need for increased clinician awareness. Furthermore, developing standardized clinical diagnostic criteria for RSV infections, similar to those for influenza-like illnesses, could be beneficial for early diagnosis and treatment.

## Figures and Tables

**Table 1 pathogens-14-00543-t001:** Baseline characteristics of patients with severe acute respiratory syndrome coronavirus 2 (SARS-CoV-2), influenza virus (INFV), or respiratory syncytial virus (RSV) infection.

Variable	SARS-CoV-2 (N = 1850)	INFV (N = 98)	RSV (N = 63)	*p*-Value
Age	70.2 ± 15.8	67.3 ± 17.9	68.9 ± 14.1	0.175
19–49 years	201 (10.9)	17 (17.3)	7 (11.1)	
50–64 years	365 (19.7)	17 (17.3)	12(19.0)	
≥65 years	1284 (69.4)	64 (65.3)	44 (69.8)	
Sex, male	1066 (57.6)	49 (50.0)	38 (60.3)	0.294
Risk group				0.377
Low risk	368 (19.9)	22 (22.4)	18 (28.6)	
Moderate risk	956 (51.7)	52 (53.1)	38 (60.3)	
High risk	525 (28.4)	24 (24.5)	7 (11.1)	
Underlying diseases				
Hypertension	1107 (59.8)	58 (59.2)	25 (39.7)	0.105
Diabetes mellitus	758 (40.1)	36 (36.7)	22 (34.9)	0.985
Cardiovascular diseases	396 (21.4)	20 (20.4)	8 (12.7)	0.575
Cerebrovascular diseases	338 (18.3)	28 (28.6)	9 (14.3)	0.116
Neuromuscular diseases	123 (6.6)	10 (10.2)	10 (15.9)	0.051
COPD	78 (4.2)	6 (6.1)	6 (9.5)	0.314
Asthma	63 (3.4)	7 (7.1)	2 (3.2)	0.422
Other chronic lung diseases	66 (3.6)	8 (8.2)	8 (12.7)	0.002 *
History of tuberculosis	75 (4.1)	2 (2.0)	4 (6.3)	0.741
Chronic renal diseases	261 (14.1)	9 (9.2)	9 (14.3)	0.737
Chronic liver diseases	87 (4.7)	2 (2.0)	3 (4.8)	0.808
Solid cancer	395 (21.3)	13 (13.3)	13 (20.6)	0.437
Hematology malignancy	78 (4.2)	4 (4.1)	6 (9.5)	0.637
Bone marrow transplant	6 (0.3)	1 (1.0)	3 (4.8)	<0.001 *
Solid organ transplant	12 (0.6)	1 (1.0)	0 (0.0)	0.992
Autoimmune diseases	36 (1.9)	5 (5.1)	0 (0.0)	0.619
Immunosuppressants user	125 (6.8)	9 (9.2)	5 (7.9)	0.903
HIV	5 (0.3)	0 (0.0)	0 (0.0)	0.971
Symptom				
None	272 (14.7)	1 (1.0)	3 (4.8)	<0.001 *
Fever	1030 (55.6)	76 (77.6)	37 (58.7)	<0.001 *
Cough	581 (31.4)	51 (52.0)	29 (46.0)	<0.001 *
Sputum	523 (28.3)	45 (45.9)	34 (54.0)	<0.001 *
Sore throat	176 (9.5)	11 (11.2)	2 (3.2)	0.484
Rhinorrhea	116 (6.3)	12 (12.2)	6 (9.5)	0.174
Dyspnea	516 (27.9)	35 (35.7)	40 (63.5)	<0.001 *
Admission route				0.192
OPD	342 (18.5)	15 (15.3)	16 (25.4)	
ER	1439 (77.7)	77 (78.6)	47 (74.6)	
Nosocomial infection	69 (3.7)	6 (6.1)	0 (0.0)	
Laboratory findings				
Total WBC counts	8.55 ± 7.6	8.82 ± 8.1	9.85 ± 5.5	0.390
AST	48.4 ± 92.5	48.3 ± 71.2	36.9 ± 25.3	0.618
ALT	33.0 ± 58.4	30.8 ± 38.7	31.1 ± 31.3	0.910
LDH	329.6 ± 291.5	327.9 ± 192.5	366.1 ± 196.6	0.722
CPK	237.1 ± 657.4	248.5 ± 472.0	149.9 ± 197.9	0.663
D-dimer	3.1 ± 5.9	1.8 ± 2.0	2.3 ± 2.2	0.245
Pro-BNP	2659.9 ± 6779.1	3105.7 ± 7023.7	2638.6 ± 4975.8	0.909
ESR	41.7 ± 31.2	39.13 ± 34.3	58.30 ± 35.4	0.006 *
CRP	50.7 ± 71.5	95.45 ± 92.2	108.06 ± 103.0	<0.001 *
Radiologic findings (CXR)				
With pneumonia	636 (34.4)	42 (42.9)	42 (66.7)	<0.001 *
Vaccination				
Influenza vaccination	818 (44.2)	46 (46.9)	38 (60.3)	0.157
COVID-19 vaccination	1537 (83.0)	87 (88.8)	59 (93.7)	0.138
Pneumococcal vaccination	1044 (56.4)	55 (56.1)	46 (73.0)	0.315

Data are presented as the mean ± standard deviation or No (%). COPD, chronic obstructive pulmonary disease; HIV, human immunodeficiency virus; OPD, outpatient department; ER, emergency room; WBC, white blood cell; AST, aspartate aminotransferase; ALT, alanine aminotransferase; LDH, lactate dehydrogenase; CPK, creatine phosphokinase; pro-BNP, pro-brain natriuretic peptide; ESR, erythrocyte sedimentation rate; CRP, c-reactive protein; CXR, chest x-ray; COVID-19, coronavirus disease 2019. * *p* < 0.05.

**Table 2 pathogens-14-00543-t002:** Comparison of clinical outcomes among patients with severe acute respiratory syndrome coronavirus 2 (SARS-CoV-2), influenza virus (INFV), or respiratory syncytial virus (RSV) infection.

Outcomes	SARS-CoV-2 (N = 1850)	INFV (N = 98)	RSV (N = 63)	*p*-Value
Cases with pneumonia	39 (39.8)	459 (24.8)	44 (69.8)	<0.001 *
ICU admission	19 (19.4)	406 (21.9)	16 (25.0)	0.925
ARDS	4 (4.1)	49 (2.6)	2 (3.1)	0.682
Mortality	7 (7.1)	189 (10.2)	7 (10.9)	0.594
Days of hospitalization	19.2 ± 26.7	14.3 ± 17.6	16.5 ± 11.9	0.141

Data are presented as the mean ± standard deviation or No (%). ICU, intensive care unit; ARDS, acute respiratory distress syndrome. * *p* < 0.05.

**Table 3 pathogens-14-00543-t003:** Risk of pneumonia development by underlying medical conditions among patients with severe acute respiratory syndrome coronavirus 2 (SARS-CoV-2), influenza virus (INFV), or respiratory syncytial virus (RSV) infection.

Underlying Medical Conditions	SARS-CoV-2 (N = 1850)			INFV (N = 98)			RSV (N = 63)		
	No (%)	OR (95% CI)	*p*-Value	No (%)	OR (95% CI)	*p*-Value	No (%)	OR (95% CI)	*p*-Value
Low risk	65/368 (17.7)	reference		5/22 (22.7)	reference		5/7 (71.4)	reference	
Moderate risk	**269/956 (28.1)**	**1.47 (1.07–2.01)**	**0.017**	25/52 (48.1)	2.47 (0.66–9.20)	0.178	27/38 (71.1)	1.08 (0.17–6.90)	0.938
1 chronic medical condition	**121/450 (26.9)**	**1.46 (1.03–2.08)**	**0.034**	10/21 (47.6)	2.55 (0.60–10.85)	0.205	16/21 (76.2)	1.40 (0.19–10.23)	0.740
2 chronic medical conditions	**93/319 (29.2)**	**1.54 (1.06–2.23)**	**0.024**	8/17 (47.1)	2.49 (0.51–12.17)	0.261	7/10 (70.0)	1.02 (0.11–9.20)	0.990
≥3 chronic medical conditions	55/187 (29.4)	1.37 (0.89–2.10)	0.154	7/14 (50.0)	2.32 (0.47–11.58)	0.304	4/7 (57.1)	0.56 (0.06–5.58)	0.618
High risk	124/525 (23.6)	1.37 (0.90–1.93)	0.074	9/24 (37.5)	1.83 (0.47–7.22)	0.387	12/18 (66.7)	0.81 (0.11–5.83)	0.838

Results with *p* < 0.05 are bolded. OR, odds ratio; CI, confidence interval.

**Table 4 pathogens-14-00543-t004:** Risk of admission to intensive care unit by underlying medical conditions among patients with severe acute respiratory syndrome coronavirus 2 (SARS-CoV-2), influenza virus (INFV), or respiratory syncytial virus (RSV) infection.

Underlying Medical Conditions	SARS-CoV-2 (N = 1850)			INFV (N = 98)			RSV (N = 63)		
	No, (%)	OR (95% CI)	*p*-Value	No, (%)	OR (95% CI)	*p*-Value	No, (%)	OR (95% CI)	*p*-Value
Low risk	74/368 (20.1)	reference		3/22 (13.6)	reference		1/7 (14.3)	reference	
Moderate risk	250/706 (26.2)	1.31 (0.97–1.77)	0.810	8/52 (15.4)	0.46 (0.09–2.42)	0.362	10/38 (26.3)	3.10 (0.28–34.68)	0.359
1 chronic medical condition	98/450 (21.8)	1.07 (0.76–1.51)	0.713	3/21 (14.3)	0.46 (0.07–10.09)	0.423	2/21 (9.5)	1.00 (0.24–38.00)	0.998
2 chronic medical conditions	86/319 (27.0)	1.38 (0.96–1.51)	0.086	2/17 (11.8)	0.30 (0.04–2.52)	0.269	3/10 (30.0)	4.55 (0.28–73.58)	0.286
≥3 chronic medical conditions	**66/187 (35.3)**	**1.38 (0.96–1.99)**	**0.001**	3/14 (21.4)	0.67 (0.10–4.63)	0.686	**5/7 (71.4)**	**28.23 (1.38–576.83)**	**0.030**
High risk	82/443 (15.6)	0.71 (0.50–1.02)	0.060	8/24 (33.3)	2.10 (0.44–10.09)	0.355	5/18 (27.8)	2.64 (0.22–32.05)	0.447

Results with *p* < 0.05 are bolded. OR, odds ratio; CI, confidence interval.

**Table 5 pathogens-14-00543-t005:** Risk of in-hospital mortality by underlying medical conditions among patients with severe acute respiratory syndrome coronavirus 2 (SARS-CoV-2), influenza virus (INFV), or respiratory syncytial virus (RSV) infection.

Underlying Medical Conditions	SARS-CoV-2 (N = 1850)			INFV			RSV		
	No, (%)	OR (95% CI)	*p*-Value	No, (%)	OR (95% CI)	*p*-Value	No, (%)	OR (95% CI)	*p*-Value
Low risk	33/368 (9.0)	reference		1/22 (4.5)	reference		1/7 (14.3)	reference	
Moderate risk	86/956 (9.0)	0.73 (0.97–2.42)	0.155	1/52 (1.9)	0.32 (0.01–7.12)	0.470	2/38 (5.3)	0.50 (0.03–7.66)	0.616
1 chronic medical condition	32/450 (7.1)	0.60 (0.38–1.02)	0.058	0/21 (0.0)	N/A	N/A	0/21 (0.0)	N/A	N/A
2 chronic medical conditions	29/319 (9.1)	0.74 (0.43–1.27)	0.280	0/17 (0.0)	N/A	N/A	0/10 (0.0)	N/A	N/A
≥3 chronic medical conditions	25/187 (13.4)	0.98 (0.55–1.74)	0.940	1/14 (7.1)	1.11 (0.05–25.64)	0.950	2/7 (28.6)	4.19 (0.20–87.26)	0.355
High risk	70/525 (13.3)	1.53 (0.97–2.42)	0.065	5/24 (20.8)	4.62 (0.44–48.62)	0.203	4/18 (22.2)	2.16 (0.16–28.32)	0.558

OR, odds ratio; CI, confidence interval; N/A, non-applicable.

## Data Availability

The datasets generated and/or analyzed during the current study are available from the corresponding author upon reasonable request.

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
