# Peer review of "Severity of Respiratory Viral Diseases and the Impacts of Underlying Medical Conditions During the Omicron Subvariant Dominant Epidemics—A Comparative Study of SARS-CoV-2, Influenza Virus and Respiratory Syncytial Virus"

_pathogens, 2025, doi:10.3390/pathogens14060543_

Round 1
Reviewer 1 Report
Comments and Suggestions for Authors Pathogens-3597393 This work aims to compare the risk factors and disease outcomes of SARS-CoV-2, RSV, and IV through a retrospective analysis. Overall, the work has multiple deficiencies, and the study include very few samples for RSV and IV compared to SARS-CoV-2 to arrive to meaningful conclusions. Specific comments are as follows.- Line 57. It should include the abbreviation for influenza virus.
- Line 73, line 81. Should include SARS-CoV-2, no COVID-19.
- Line 79. It needs to be indicated where the hospitals were located. In Korea? Which part?
- Line 80, line 138. The authors need to indicate "We...when the Omicron subvariants were dominant". However, they should indicate which virus they are referring to.
- Line 137. A major problem with this work is that the comparative analysis is disproportionately assessed for RSV and IV. Very few samples for those 2 viruses were analyzed to be compared for % of incidence/outcomes.
- Line 153. If the group also included patients that were asymptomatic but infected, the group should be SARS-CoV-2 positive, no COVID-19 positive.
- line 163-164 and Table 1. The percentage for RSV is inaccurate and does not match the numbers indicated in the Table. It is not clear how the % for SARS-CoV-2 and IV were calculated based on the number of samples analyzed. It looks like the data are wrong.
- Table 3. The presentation of the information is confusing and difficult to read. Moreover, no results with p<0.05 are in bold, as indicated in the note (line 177).
- Line 208. COVID-19 is not a virus.
- line 276. The authors indicate "...the severity of COVID-19 is higher in older patients and with multiple underlying medical conditions". However, it is not clear older compared to what or what age range they are referring to.
Author Response
Major Comments 1) Line 57. It should include the abbreviation for influenza virus.
Answer) We sincerely appreciated for your time and valuable comments. We have revised the manuscript to include the abbreviation for influenza virus (IFV) at its first mention, as appropriate.
Line 59-62: Influenza virus (IFV), respiratory syncytial virus (RSV), and severe acute respiratory syndrome coronavirus 2 (SARS-CoV-2) are circulating concurrently, leading to high morbidity and mortality.
Major Comments 2) Line 73, line 81. Should include SARS-CoV-2, no COVID-19.
Answer) ) Thank you for your kind comments. As suggested, we have revised the text in lines 73 and 81 to refer to SARS-CoV-2 instead of COVID-19, to ensure virologically accurate terminology.
Line 75-77: This study aims to compare the clinical outcomes and risk factors of severe infections among these three representative respiratory viral infections (SARS-CoV-2, influenza, and RSV).
Line 85-86: We included all hospitalized patients aged 19 years or older with laboratory-confirmed SARS-CoV-2 COVID-19, IFV, or RSV infections during this period.
Major Comments 3) Line 79. It needs to be indicated where the hospitals were located. In Korea? Which part?
Answer) As we fully respect your recommendation, we have revised the sentence to specify the geographic location of the participating hospitals.
Line 80-82: We conducted a multicenter, retrospective cohort study involving eight hospitals located in various regions (Seoul, Suwon, Ansan, Incheon, Wonju, Daegu, Pusan, Gwangju, and Chungbuk) of South Korea, all participating in the Hospital-based Influenza Morbidity and Mortality surveillance system
Major Comments 4) Line 80, line 138. The authors need to indicate "We...when the Omicron subvariants were dominant". However, they should indicate which virus they are referring to.
Answer) Thank you for your nice comment. We have revised the text in lines 80 and 138 to specify that the Omicron subvariants refer to those of SARS-CoV-2, for clarity and accuracy.
Line 83-85: The study spanned from September 1, 2022, to August 31, 2023, during periods when the SARS-CoV-2 Omicron subvariants circulated dominantly.
Line 149-152: A total of 2,011 hospitalized patients were included in the analysis across eight hospitals from September 1, 2022, to August 31, 2023, during periods when the SARS-CoV-2 Omicron subvariants circulated dominantly.
Major Comments 5) Line 137. A major problem with this work is that the comparative analysis is disproportionately assessed for RSV and IV. Very few samples for those 2 viruses were analyzed to be compared for % of incidence/outcomes.
Answer) We agree that the number of cases for RSV and IV was relatively small compared to SARS-CoV-2, which limits the robustness of the comparative analysis. We have now addressed this issue in the revised manuscript by explicitly stating it as a limitation in the discussion section, and we have added a note of caution in interpreting the corresponding results.
Line 306-311: Secondly, the study was limited to a single season shortly following the COVID-19 pandemic. This limited timeframe, coupled with a smaller number of IFV and RSV cases, posed challenging to achieve statistical significance for some comparisons. Therefore, the comparative analysis of incidence and clinical outcomes between SARS-CoV-2, IFV, and RSV should be interpreted with caution due to the disproportionate distribution of cases.
Major Comments 6) Line 153. If the group also included patients that were asymptomatic but infected, the group should be SARS-CoV-2 positive, no COVID-19 positive.
Answer) As you recommended, we have revised the text to refer to SARS-CoV-2 positive patients rather than COVID-19 positive patients.
Line 168-169: Patients with SARS-CoV-2 infection were prone to be asymptomatic and experienced cough and sputum less frequently compared to those with influenza or RSV.
Major Comments 7) line 163-164 and Table 1. The percentage for RSV is inaccurate and does not match the numbers indicated in the Table. It is not clear how the % for SARS-CoV-2 and IFV were calculated based on the number of samples analyzed. It looks like the data are wrong.
Answer) Thank you for your kind comment. We re-examined the data and identified an error in the reported percentage for RSV. The correct value has now been updated in both the manuscript and Table 1. Additionally, we have clarified that all percentages for SARS-CoV-2, IFV, and RSV were calculated based on the number of each virus-positive patients. We have modified the sentence to explicitly indicate the numerators and denominators for clarity and consistency.
Line 169-172: Among patients positive for each virus, those with IFV exhibited a higher prevalence of fever (76/98, 77.6%, p<0.001), whereas dyspnea was more common among patients with RSV infection (40/63, 63.562.5%, p<0.001; Table 1).
Major Comments 8) Table 3. The presentation of the information is confusing and difficult to read. Moreover, no results with p<0.05 are in bold, as indicated in the note (line 177).
Answer) Thank you for your valuable feedback. We have identified an error in bolding statistically significant results and this has been corrected.
Regarding the structure of the table, we agree that it may appear dense. However, given the number of variables and comparisons, we selected this format to present the data as clearly and efficiently as possible. Minor adjustments have been made to improve readability while maintaining the necessary level of detail.
Major Comments 9) Line 208. COVID-19 is not a virus.
Answer) As noted in the earlier context, we have revised the terminology to accurately reflect the distinction and have replaced "COVID-19" with "SARS-CoV-2" where appropriate.
Line 240-241: This study evaluated the clinical outcomes of three major respiratory viruses (SARS-CoV-2, IFV, and RSV)
Major Comments 10) line 276. The authors indicate "...the severity of COVID-19 is higher in older patients and with multiple underlying medical conditions". However, it is not clear older compared to what or what age range they are referring to.
Answer) Thank you for your comment. We have clarified that "older patients" refers to those aged 65 and older, with greater severity of COVID-19 compared to younger individuals.
Line 321-322: The severity of COVID-19 is higher in patients aged 65 and older, particularly those with multiple underlying medical conditions.
Reviewer 2 Report
Comments and Suggestions for Authors
The article is the first systematic comparison of clinical outcomes and the impact of underlying diseases on the severity of COVID-19, influenza, and RSV infections during the omicron subtype-dominated pandemic, which fills the current gap of comparative multiviral data from the transition from pandemic to endemic phase, which has some reference value. However, many areas require further and more refined explanation by the authors. For example, the following articles are for reference only:
1. The impact of inter-viral interactions (e.g. co-infection) or novel biomarkers (e.g. cytokine storm) was not explored in depth and the innovation was limited to the description of clinical phenotypes, which is recommended to be completed.
2. although it is clear that RSV infection has the highest incidence of pneumonia in elderly hospitalised patients (65.6%), the authors chose an unbalanced sample size (COVID-19: 1850 cases vs. RSV: 63 cases), the statistical power of the RSV and influenza groups was insufficient, and extrapolation of the conclusions was limited.
3. The retrospective design, which relies too heavily on electronic medical record data, carries a risk of under-reporting, and the statistical methodology was not corrected for multiple comparisons, which may lead to the possibility that some of the results are false positives.
4. The study population consisted mainly of elderly Korean inpatients (69.4% aged ≥65 years) and did not include outpatients or patients with minor illnesses, and the findings may not be applicable to other populations (e.g., children or non-East Asian populations).
5. Some of the references cited are out of date and it is recommended that they be updated, preferably with the latest reports within the last five years.
6. Off-topic, I checked the replication rate of the article and it was as high as 29%, which is unacceptable and needs to be further optimised and adjusted.
Author Response
Major Comments 1) The impact of inter-viral interactions (e.g. co-infection) or novel biomarkers (e.g. cytokine storm) was not explored in depth and the innovation was limited to the description of clinical phenotypes, which is recommended to be completed.
Answer) We sincerely appreciated your time and thoughtful feedback of the manuscript. We agree with your opinion and have added this limitation in the discussion.
Line 318-320: Finally, the study did not explore inter-viral interactions, and future research should investigate co-infections and biomarkers to better understand disease severity and prognosis.
Major Comments 2) Although it is clear that RSV infection has the highest incidence of pneumonia in elderly hospitalized patients (65.6%), the authors chose an unbalanced sample size (COVID-19: 1850 cases vs. RSV: 63 cases), the statistical power of the RSV and influenza groups was insufficient, and extrapolation of the conclusions was limited.
Answer) We understand and acknowledge this limitation. As mentioned in the Discussion section, RSV testing may be underutilized in clinical practice due to the lack of specific treatments, which could lead to an underestimation of its true burden. Nevertheless, we believe that our findings highlight the clinical importance of RSV and may contribute to raising awareness and guiding future research in this area.
Line 306-309: Secondly, the study was limited to a single season shortly following the COVID-19 pandemic. This limited timeframe, coupled with a smaller number of influenza and RSV cases, posed challenging to achieve statistical significance for some comparisons. Therefore, the comparative analysis of incidence and clinical outcomes between SARS-CoV-2, influenza, and RSV should be interpreted with caution due to the disproportionate distribution of cases.
Major Comments 3) The retrospective design, which relies too heavily on electronic medical record data, carries a risk of under-reporting, and the statistical methodology was not corrected for multiple comparisons, which may lead to the possibility that some of the results are false positives.
Answer) We acknowledge the limitations associated with the retrospective design and the reliance on electronic medical records, and we have addressed these concerns in the limitation section. To mitigate the possibility of false positives arising from multiple comparisons, we performed post-hoc analyses.
Line 125-128: To control for false positives, post-hoc analyses were conducted following significant findings. For chi-square tests, adjusted residuals were used to identify cell-level differences, and for ANOVA, Bonferroni correction was applied.
Line 183-185: Both adjusted residuals analysis (using a significance threshold of ±1.96) and Bonferroni correction consistently confirmed the presence of significant cell-level differences across all relevant comparisons.
Line 302-304: First, the retrospective design may have led to an underestimation of the disease burden associated with RSV infections, as mild cases could have been missed or underreported.
Major Comments 4) The study population consisted mainly of elderly Korean inpatients (69.4% aged ≥65 years) and did not include outpatients or patients with minor illnesses, and the findings may not be applicable to other populations (e.g., children or non-East Asian populations).
Answer) We sincerely appreciate your valuable feedback and have revised the manuscript to clarify this limitation.
Line 312-314: Third, as mentioned above, most of our study population consisted of elderly hospitalized patients (69.4% aged ≥65 years), which may limit the generalizability of our findings to other populations
Major Comments 5) Some of the references cited are out of date and it is recommended that they be updated, preferably with the latest reports within the last five years.
Answer) Thank you for your helpful comment. As suggested, we have updated the Discussion section by incorporating the literature review and added the most recent references.
Line 272-276: Additionally, a retrospective study conducted in China reported lower survival rates following RSV infection among elderly patients and those with congestive heart failure or chronic obstructive pulmonary disease [19]. Similarly, a study from Croatia found that chronic conditions, particularly cardiovascular and respiratory diseases, were strongly associated with higher complication rates in RSV-infected patients [20].
Major Comments 6) Off-topic, I checked the replication rate of the article and it was as high as 29%, which is unacceptable and needs to be further optimized and adjusted in the analysis.
Answer) As we fully respect your recommendation, we have carefully reviewed and revised the methods section to reduce overlap with previous literature. While certain standard descriptions and terminologies are commonly used in this field and are necessary for clarity and accuracy, we have rephrased and reorganized the relevant sentences wherever possible to improve originality. We hope these changes address your concerns.
Round 2
Reviewer 1 Report
Comments and Suggestions for Authors
Pathogens-3597393
This is a revised version of the manuscript, in which the authors answered most of the comments. However, the main weakness of this manuscript remains the small number of samples analyzed for RSV, which makes this manuscript focused on SARS-CoV-2 in a special issue of RSV. Specific comments are as follows.
- Line 34. The acronym used for the influenza virus is incorrect. According to the official taxa, IFV corresponds to Infectious Flacherie Virus, not influenza. That needs to be corrected accordingly throughout the text.
- Line 151. A major problem with this work is that the comparative analysis is disproportionately assessed for RSV. Very few samples were analyzed to be compared for % of incidence/outcomes.
- Overall, the emphasis of this manuscript, including the title, is on the Omicron subvariant of SARS-CoV-2. Nevertheless, the special issue emphasizes the epidemiology of RSV and the potential impact of vaccination.
- Line 323. Should read: “…in patients aged 65 and older, particularly…’ not order.
- The manuscript will benefit from an additional round of editing.
Author Response
Major Comments 1) Line 34. The acronym used for the influenza virus is incorrect. According to the official taxa, IFV corresponds to Infectious Flacherie Virus, not influenza. That needs to be corrected accordingly throughout the text.
Answer) As we fully respect your opinion, we have revised the text accordingly by replacing all instances of "IFV" with the term "INFV" throughout the manuscript to avoid confusion and ensure taxonomic accuracy. Although we considered using the abbreviation 'IV' for the influenza virus, we decided on 'INFV' instead, as 'IV' is commonly understood to mean 'intravenous.' We appreciate your understanding.
Major Comments 2) Line 151. A major problem with this work is that the comparative analysis is disproportionately assessed for RSV. Very few samples were analyzed to be compared for % of incidence/outcomes.
Answer) Thank you for your thoughtful comment. We understand your concern regarding the limited sample size for RSV analysis. It is likely that RSV cases were underestimated, and we agree that further research with larger sample sizes is needed. We have clarified this point further in the limitations section of the manuscript.
Line 311-315: The limited timeframe and relatively few INFV and RSV cases reduced the statistical power of certain comparisons. Consequently, interpretations of incidence and clinical outcomes across SARS-CoV-2, INFV, and RSV should be approached with caution, particularly given the disproportionately low number of RSV cases, which may compromise the robustness of the findings.
Major Comments 3) Overall, the emphasis of this manuscript, including the title, is on the Omicron subvariant of SARS-CoV-2. Nevertheless, the special issue emphasizes the epidemiology of RSV and the potential impact of vaccination.
Answer) Thank you for your valuable comment. We acknowledge that the primary focus of our manuscript is on the Omicron subvariant of SARS-CoV-2. However, we have made efforts to align with the aims of the special issue by including comparative analyses with RSV, particularly in terms of epidemiological trends and clinical characteristics. Although there are clear differences in sample sizes between respiratory viruses, this study specifically focuses on the underestimation of RSV, particularly in adults. We have addressed this issue in the discussion and emphasize the need for further research to improve diagnostic rates. We hope this manuscript contributes to raising awareness of the importance of such future studies.
Line 304-307: In particular, mild RSV infections in adults are often underestimated, underscoring the need for future research to develop and validate a standardized case definition that can facilitate more accurate diagnosis and monitoring, especially in outpatient and community settings
Major Comments 4) Line 323. Should read: “…in patients aged 65 and older, particularly…’ not order.
Answer) Thank you for your careful comment. We have made the correction by replacing "order" with "older" as suggested.
Line 325-326: This study highlights that the severity of SARS-CoV-2 is higher in patients aged 65 and older …
Major Comments 5) The manuscript will benefit from an additional round of editing.
Answer) We appreciate your suggestion and will carefully review the manuscript for further editing to improve its clarity and quality.
Reviewer 2 Report
Comments and Suggestions for Authors
The author's response letter is scientifically sound and reasonable. The revised paper has largely addressed the main issues I raised, with supplementary materials supporting the research conclusions. I recommend making the following minor adjustments before acceptance: further emphasize future research directions in the discussion section.
Author Response
The author's response letter is scientifically sound and reasonable. The revised paper has largely addressed the main issues I raised, with supplementary materials supporting the research conclusions. I recommend making the following minor adjustments before acceptance: further emphasize future research directions in the discussion section.
Answer) Thank you for your valuable suggestion. We have revised the paragraph of the limitation section to more clearly outline future research directions.
Line 304-307: In particular, mild RSV infections in adults are often underestimated, underscoring the need for future research to develop and validate a standardized case definition that can facilitate more accurate diagnosis and monitoring, especially in outpatient and community settings.